# Early Childhood Junk Food Consumption, Severe Dental Caries, and Undernutrition: A Mixed-Methods Study from Mumbai, India

**DOI:** 10.3390/ijerph17228629

**Published:** 2020-11-20

**Authors:** Priyanka Athavale, Nehaa Khadka, Shampa Roy, Piyasree Mukherjee, Deepika Chandra Mohan, Bathsheba (Bethy) Turton, Karen Sokal-Gutierrez

**Affiliations:** 1School of Public Health, University of California, Berkeley, CA 94704, USA; nehaakhadka@gmail.com (N.K.); s.roy.0891@gmail.com (S.R.); deepikachandra@berkeley.edu (D.C.M.); ksokalg@berkeley.edu (K.S.-G.); 2School of Medicine, University of California, San Francisco, CA 94143, USA; 3Fielding School of Public Health, Department of Epidemiology, University of California, Los Angeles, CA 90095, USA; 4Swasti Health Catalyst, Mumbai 560094, India; piyasree@gmail.com; 5University of Puthisastra, Phnom Penh 12211, Cambodia; bethy.turton@gmail.com

**Keywords:** India, nutrition transition, junk food, sugar-sweetened beverages, early childhood caries, ECC, mouth pain, undernutrition, malnutrition, mixed-methods study

## Abstract

In India, globalization has caused a nutrition transition from home-cooked foods to processed sugary snacks and drinks, contributing to increased early childhood caries (ECC). This mixed-methods study describes risk factors for ECC and associations with undernutrition in low-income communities in Mumbai. Interviews with mothers of 959 children, ages six-months through six-years, addressed maternal-child nutrition and oral health, and children received dental exams and anthropometric assessments. Focus groups with community health workers and mothers explored experiences and perceptions of oral health, nutrition, and ECC. Descriptive and logistic regression analyses of quantitative data, and content analysis of qualitative data were performed. Eighty percent of children lived 5 min from a junk-food store, over 50% consumed junk-food and sugary tea daily, 50% experienced ECC, 19% had severe deep tooth decay, 27% experienced mouth pain, and 56% experienced chronic and/or acute malnutrition. In children ages 3–6, each additional tooth with deep decay was associated with increased odds of undernutrition (Odds Ratio [OR] 1.10, Confidence Interval [CI] 1.02–1.21). Focus groups identified the junk-food environment, busy family life, and limited dental care as contributors to ECC. Policy interventions include limits on junk-food marketing and incorporating oral health services and counseling on junk-food/sugary drinks into maternal–child health programs.

## 1. Introduction

The nutrition transition in low-middle-income countries (LMICs) is leading to increased rates of non-communicable diseases (NCDs) such as obesity, diabetes, cardiovascular disease, and oral diseases [1,2,3,4,5]. Economic development, globalization, and urbanization have led populations to shift from traditional minimally-processed diets rich in staple foods of vegetable origin to diets high in meat, ultra-processed snack foods high in sugar, fat, and salt, and sugar-sweetened beverages [1,2,3,6]. Large-scale studies have shown consumption has dramatically increased in LMICs, particularly in Asia, with booming fast food and beverage industries [7,8].

Frequent consumption of sugar—in sugary liquids in the baby bottle, carbohydrate-dense and sugary snack foods and beverages—has been identified as the main cause of early childhood caries (ECC) in children under age six [4]. Other risk factors for ECC include insufficient tooth brushing, inadequate fluoride, poor access to dental care, and family socioeconomic and educational factors [9]. Additionally, several studies have documented an association between ECC and child malnutrition. In high-income countries, studies have shown associations between ECC and obesity, likely due to the common dietary risk factors. In LMIC, studies have shown associations between ECC and undernutrition. These have hypothesized a bidirectional association, with undernutrition contributing to ECC due to enamel hypoplasia and poor immune defenses, and ECC contributing to undernutrition due to chronic dental infections, mouth pain, and poor nutritional intake [10,11,12,13,14].

India has been undergoing the nutrition transition over the past two decades, with food industry growth at a rate of 40% each year [15,16,17]. Increasing childhood consumption of junk food and rising rates of ECC have been documented. A systematic review estimated an overall 49.6% prevalence of ECC in children aged 2–6 years in India, with no state reporting a prevalence of less than 40% [18,19]. A cross-sectional study of 200 preschool-aged children in an urban area in Maharashtra state showed an 87.5% prevalence of caries, associated with frequent snacking habits and poor tooth brushing frequency [20]. In India, the nutrition transition has also contributed to increasing rates of obesity in middle-higher income populations, with the persistence of undernutrition in low-middle income populations, leading to a “double-burden” of obesity and undernutrition [21]. Despite India’s many interventions to combat child malnutrition, malnutrition rates have plateaued with 38% of children stunted, 36% underweight, and 26% wasted [22,23,24].

It is critical to study the relationships between junk food consumption, early childhood caries, and child malnutrition in India to design interventions addressing nutrition and oral health practices to improve children’s health. This study aims to use a mixed-methods approach to describe the risk factors for ECC and associations with malnutrition in a convenience sample of children and families from low-income urban communities in Mumbai, India.

## 2. Materials and Methods

### 2.1. Study Design and Population

This was a mixed-methods study of children’s nutrition and oral health in a convenience sample of families living in five urban slum communities in Mumbai, India. Quantitative data was collected by mother/caregiver interviews and child dental and anthropometric exams; qualitative data was collected by focus groups with mothers/caregivers and health workers. Study inclusion criteria consisted of mother/caregivers with children from 6 months through 6 years of age, with no exclusions for medical reasons. The urban slum communities are informal housing settlements with up to 1 million people living in densely-packed areas with poor hygienic conditions. Most households included two parents, 1–3 children, some with grandparents, aunts, uncles and other family members, living within 50–100 sq ft. The study was conducted in partnership with three local Indian non-governmental, non-profit organizations (NGOs) working to promote children’s health and development in Mumbai slums: Foundation for Mother and Child Health (FMCH), Community Outreach Programme (CORP), and Reality Gives (RG). The study received ethical approval by the University of California Berkeley Committee for the Protection of Human Subjects (#2012-11-4798), University of California San Francisco by the UC Reliance System (#369-4), and the directorship of FMCH, CORP, and RG. This research study was performed in compliance with the Helsinki Declaration.

### 2.2. Quantitative Methods

#### 2.2.1. Data Collection/Assessment Instruments

From December 2012 through December 2015, the three NGO staff used word-of-mouth to recruit a convenience sample of mothers/caregivers of children aged 6 months through 6 years in the five urban slum communities where they worked. All families with children in this age group were invited to participate. Annual “India Smiles” dental camps involved baseline data collection followed by free oral health and nutrition education for children and adults, toothbrushes, and fluoride toothpaste for all family members, fluoride varnish applications for children, individualized dental advice, and referrals to dental care as needed. Trained NGO staff verbally explained the study protocol to participating mothers/caregivers and children in their local language (Marathi or Hindi), and obtained informed consent through written signatures and/or verbal assents. Data collection included:(1)Nutrition and Oral Health Survey: Trained NGO staff and university student volunteers fluent in the local languages interviewed each mother/caregiver in her preferred language regarding maternal-child nutrition and oral health. The survey instrument was modified from the World Health Organization (WHO) oral health survey [25]. This survey consisted of 50 questions, with 22 questions regarding maternal nutrition and oral health knowledge and practices, and 28 questions regarding children’s nutrition and oral health.(2)Dental Screening Exams: Licensed Indian dentists performed child dental screening exams using visual inspection with light and mirror. They recorded decayed, missing and filled teeth, and estimated depth of cavitation into the enamel (d1), dentin (d2), or pulp (d3), based on WHO standards [25]. The dentists calibrated their exams by independently and then jointly examining 5 children and agreeing on findings.(3)Anthropometric measures: Trained community health workers and university student volunteers measured each child’s weight and height or length, without shoes and in light clothing, using a digital scale and stadiometer (Seca, Chino, CA, USA), according to WHO standards [26].

#### 2.2.2. Statistical Analysis

Data were entered into SAS version 9.4 (SAS Institute., Cary, NC, USA). A “baseline” dataset was created with the first visits for each mother-child pair at any dental camp from 2012 to 2015. There were 250 cases who were siblings, and maternal data were entered separately for each case. Missing data were assumed to be missing at random and were ignored in the final analysis. Descriptive, bivariate, and logistic regression analyses were performed as informed by the conceptual model presented in Figure 1. Junk food consumption was estimated according to the number of exposures per day, and divided into tertiles of exposure frequency. Anthropometric data on child weight, height/length and age were entered into WHO AnthroPlus Software 3.2.2 (WHO, Geneva, Switzerland). to determine each child’s standardized height-for-age (HAZ), weight-for-age (WAZ), and Body Mass Index-for-age (BAZ) to assess for stunting, underweight and wasting, respectively. A child was categorized as “undernourished” for a Z-score less than –2 for HAZ, WAZ, or BAZ. Bivariate associations were conducted between junk food consumption tertiles, severe dental caries indicators (d3, mouth pain), and undernutrition indicators (BAZ, HAZ, WAZ, any malnutrition) (see Appendix A). Simple binomial logistic regression modelling was used to explore the hypothesis that there was an association between severe dental caries (as defined by deep decay) and undernutrition, examining two different age-groups—children under age 3 (with shorter exposure duration, and erupting primary teeth) and children age 3 through 6 (with longer exposure duration and completely-erupted primary teeth). The number of teeth with deep decay lesions (d3) was entered into the model as a continuous variable, and the model controlled for the mother’s education level in years of schooling, child gender, and junk food consumption tertiles.

### 2.3. Qualitative Methods

#### Focus Groups—Participant Recruitment, Data Collection and Analysis

In conjunction with the Foundation for Mother and Child Health (FMCH), four focus groups were conducted, with 42 health workers and mothers/caregivers of young children living in two urban slum communities in Mumbai, India. FMCH staff recruited participants considered spokespeople for their community and fluent in Hindi which was the language used for focus group discussions. Two focus groups were conducted in 2012 and two in 2015. Focus groups were led by a trained facilitator and a note-taker. Participants provided informed consent and agreement of confidentiality. The discussion followed a semi-structured focus group guide addressing participants’ oral health practices and experiences, perceived barriers and facilitators to good oral health, and recommendations for improving children’s oral health. The focus group discussions lasted 60–90 min, were audio-recorded, and subsequently transcribed from Hindi to English. Transcripts were analyzed using content analysis, with initial independent review and coding by two researchers (D.C.M., K.S.-G.) to identify preliminary themes, followed by discussion and agreement between coders on themes and subthemes using inductive and deductive methods [27]. Major themes, subthemes and illustrative quotes were identified.

## 3. Results

### 3.1. Quantitative Results

The study sample included a total of 959 children (Table 1). The mean age of the children was 3.7 years. Mothers had a mean age of 27 years, with an average of 6 years of education. The average household size was six individuals, including 2–3 children. The majority of families had access to electricity, potable water, and cooking fuel other than wood.

Access to junk food was measured by the estimated time to walk from home to a store selling junk food. Most families (81%) lived within a 5-min walk to the store.

#### 3.1.1. Maternal Oral Health and Nutrition Knowledge, Practices and Status

Mothers’ knowledge about oral health was assessed with an open-ended question, “What do you think causes child tooth decay?” Most mothers (75%) knew that eating sweets caused caries, but less than a third of mothers (30%) knew that not brushing teeth contributed to caries, and a very small proportion knew that drinking soda/juice and bottle-feeding contributed to caries (Table 2).

Most mothers (87%) reported drinking tea with sugar every day, but a low proportion reported daily consumption of chips or biscuits, sweets, and soda, and one-fifth of mothers consumed plain milk daily.

Nearly all mothers reported having their own toothbrush; however, only 42% had ever been to the dentist, compared to nearly all mothers who had received prenatal care.

Overall, 39% of mothers reported having dental problems during the past three months, primarily dental pain in nearly one-third of mothers (31%), and smaller proportions with decayed or loose teeth, bleeding gums, and inflammation.

#### 3.1.2. Child Dietary and Oral Health Practices

Most children (93%) were breastfed for an average of 21 months (Table 3). More than 1 in 4 children (28%) were bottle-fed, generally in addition to breastfeeding, with an average duration of 19 months. Of those that were bottle-fed, nearly half of babies (46%) were put to bed with the bottle, and 5% were fed sugary drinks in the bottle. The majority of children consumed cariogenic items daily—sweets, candy or chocolate (52%), chips and biscuits (58%), and tea with sugar (51%); the frequency of daily consumption of these items was higher in children over age three. Daily consumption of soda and juice was low. Most children (63%) were given milk daily, with a higher frequency in children under three years. Most mothers (61%) reported that they spent at least Rs 5 ($0.07) on junk food per child daily, and a greater proportion reported this higher expense for children under age three. 

In terms of oral health practices, most children reportedly had their own toothbrush and toothpaste. Nearly two-thirds of mothers (63%) reported helping their children brush frequently, and over one-third of mothers (37%) reported doing nothing to help care for their child’s teeth. Only 14% of children had been to the dentist, compared to nearly all children (98%) being up-to-date on their immunizations.

#### 3.1.3. Child Oral Health and Nutrition Status

Overall, 50% of children had evidence of tooth decay and nearly all of the tooth decay (96%) was untreated (Table 4). The number of decayed, missing or filled teeth (dmft) per child ranged from 0 to 20, with a mean dmft of 2.7 for all children, and 5.4 for the children with any decay. Approximately 1 in 4 children (23%) had 5 or more decayed teeth. Nearly 1 in 5 children (18.6%) had deep decay, with a range of up to 15 teeth with deep decay; and over one-fourth of children (27%) complained of mouth pain. Overall, 1 in 5 mothers (20%) reported considering their child’s oral health was “bad” compared to only 12% that their child’s overall health was “bad.” Overall, 56% of all children were undernourished, with 42% stunted, 36% underweight and 21% wasted.

Evidence of early childhood caries was seen in the first year of life, and increased steadily in frequency and severity by age (Figure 2). At age six, over 80% of children experienced tooth decay, the mean dmft was approximately six teeth, 50% of children had deep decay, and 54% of children complained of mouth pain.

#### 3.1.4. Association of Undernutrition with Deep Decay and Junk Food Consumption

Logistic regression analysis demonstrated that, for children age 3 through 6 years, there was a moderate association between severe tooth decay (by deep decay, d3) with increased odds of undernutrition (aOR: 1.10, 95% CI: 1.02–1.21), i.e., for each additional tooth decayed to the deep level, there was a 10% increased odds of stunting, underweight or wasting (Table 5). Frequency of junk food consumption appeared to moderate the odds of undernutrition (aOR: 0.80, 95% CI: 0.65–0.98); i.e., for each higher-frequency tertile of junk food consumption, there was a 20% lower odds of stunting, underweight, or wasting.

### 3.2. Qualitative Results

The 42 participants across 4 focus groups included 11 mothers/caregivers of young children and 5 FMCH health workers in 2012, and 14 pregnant women, and 12 mothers with infants in 2015. All participants were women over age 18, with literacy levels from illiterate to high school level. Three major themes were identified: (1) Families’ oral health knowledge, attitudes, practices, and experiences; (2) Contributors to child tooth decay; and (3) Suggestions to improve children’s oral health. Each theme had 5–6 subthemes, and representative quotes (Table 6).

#### 3.2.1. Theme 1: Families’ Oral Health Knowledge, Attitudes, Practices and Experiences

Most participants were aware of the importance of good oral health, for adults and children. They associated good oral health with having healthy gums and teeth, good breath, attractive smiles, self-esteem, good overall health and fewer illnesses. They discussed traditional practices to maintain oral health such as rinsing their mouth after meals, cleaning their teeth with their finger or traditional toothbrush (e.g., neem or lime plant stick) and toothpaste substitutes (e.g., asafetida, salt, charcoal, sand, and tobacco); and learning about modern oral hygiene techniques from family, neighbors, schools, health professionals, and advertisements on television.

Most participants reported experience with oral health problems—for themselves, adult family members and their children—including decayed teeth, dental pain, swelling, and bleeding gums. They described how oral health problems caused substantial suffering for child nutrition, health, and well-being.

#### 3.2.2. Theme 2: Contributors to Child Tooth Decay

Participants believed that child tooth decay was primarily caused by children’s easy access to inexpensive “junk food” (e.g., candy, chocolate, chips, biscuits, and sugary drinks) near their homes and their children’s schools. They said that children were very attracted to junk food and asked their parents to buy it or give them money to buy it, which most parents did; and it was challenging for them to resist social pressure and restrict their own children’s pocket money. Mothers explained that their busy family living and working conditions made it difficult to monitor and limit their children’s junk food intake. Mothers said that they prepared home-cooked meals, but it was difficult to persuade their children to eat meals. Some said that relatives pampered their children with junk food, which contributed to tooth decay and overweight. In addition, most participants acknowledged their own fondness for drinking tea with biscuits several times a day, and some reported drinking other sugar-sweetened beverages because they helped them to stay alert and active during housework.

Most participants were aware of the importance of good oral hygiene practices, particularly daily tooth brushing. However, they reported challenges in supervising their children’s tooth brushing because they were too busy with other family responsibilities and their children protested brushing because they disliked the taste of toothpaste or experienced pain with tooth brushing.

Some participants reported having visited a dentist for treatment rather than preventive services. Only one participant described excellent treatment from a dentist, and others described negative experiences, including the high cost of dental care, delayed treatment, extraction as the primary treatment, fear of pain during treatment, and perceived poor quality of care. Many preferred to use home remedies (e.g., clove, neem twigs), and sought professional dental care only if their pain was unbearable and unmanageable with home remedies.

#### 3.2.3. Theme 3: Suggestions to Improve Children’s Oral Health

Participants proposed some ideas to prevent children from eating junk food, including avoiding giving children pocket money to buy sweets, preparing a variety of foods at home, and giving children fruit instead of sweets. Participants said that receiving free oral hygiene supplies helped parents and children develop the habit of brushing together, and parents shared creative strategies to motivate their children to brush. Health workers noted that the families who regularly attended their FMCH clinics—that incorporated messages to avoid junk food and ensure daily toothbrushing—appeared healthier. One participant suggested motivating parents by demonstrating the cost savings from preventing tooth decay. Participants agreed that they needed better access to affordable dental care.

## 4. Discussion

This mixed-methods study describes the socio-environmental and behavioral context for children’s frequent junk food consumption, development of tooth decay, and associated undernutrition in low-income, urban communities in Mumbai, India. The quantitative components (surveys and exams) described daily consumption of junk food, progressing frequency, and severity of untreated ECC from infancy to age six, and the association between severe ECC and under-nutrition in children aged 3 through 6. The qualitative component (focus groups) described key challenges to ensuring good oral health and nutrition—widely-available and inexpensive junk food; parents giving children money to buy snacks, not enforcing tooth brushing, and poor access to affordable and dental services, leaving most dental disease untreated. These findings support other studies in India and LMICs examining child nutrition and oral health, and they offer additional perspective by examining these factors together, from quantitative and qualitative perspectives.

### 4.1. Junk Food Environment

An ecological model can help explain the multi-level factors affecting children’s oral health in our population [28]. Factors at both global and community level drive critical behaviors at the family and child level: primarily daily consumption of junk food. Globalization and urbanization have led to wide availability of low-cost junk foods at stores, within a 5 min walk from most homes in this study. Busy family life, and the attraction of junk food, have created social pressure for parents to give children money to buy snacks after school. Most parents spent at least Rs 5 daily per child on junk food, with the average cost of candy/sweets ranging between Rs 1–10, with over half of the children consuming candy, chocolate, biscuits, and sugary tea daily. Moreover, junk food consumption started at a very young age, with over half of children less than three years consuming chips and biscuits daily. Early and frequent exposure to sweets can develop taste preferences, addiction to sugar, and lifelong habits contributing to poor oral health and nutrition and other NCDs [29]. These findings support other studies on accessibility and consumption of junk food by very young children, and junk food marketing to attract school-aged children to purchase snacks afterschool [8,30,31,32]. These food-environment challenges highlight the need for policy initiatives such as prohibiting marketing of junk food to children, adding warning labels on unhealthy products, and taxing sugary drinks and snacks [33,34,35]. Over recent years, India instituted higher taxes on sugar-sweetened beverages to prevent obesity, which could contribute to ECC prevention as well [15,34].

### 4.2. Maternal Knowledge, Practices, and Barriers to Oral Health and Nutrition Recommendations

We found gaps in mothers’ knowledge of the causes and prevention of ECC, as well as gaps in applying their knowledge to practice. Although most mothers knew that tooth decay was caused by sweet snacks, few knew that it was caused by sweetened drinks, and most gave their children sweet snacks and sugar-sweetened tea daily. Mothers followed the traditional diet themselves, and sugary tea as the only cariogenic item that they consumed. However, they were 3–5 times as likely to give their children daily candy, chocolate, biscuits, and chips than consume it themselves. Many children live in joint-family structures, and were given sweets by extended family members (grandparents, aunts, and uncles), similar to results from another qualitative study that identified mothers-in-law as key decision-makers for children’s diets [36].

Regarding oral hygiene, most mothers were not aware that lack of tooth brushing can lead to tooth decay, over one-third of mothers reported doing nothing to care for their child’s teeth, mothers reported being too busy to brush their children’s teeth, and their children avoided brushing. These barriers highlight the need to educate parents on the importance of daily tooth brushing with fluoride toothpaste, and that children need adult help and supervision until 6–8 years of age.

One in three mothers and one in four children in our sample suffered from mouth pain—likely from untreated caries—and less than half of mothers and one in seven children had ever been to the dentist. Mothers reported a lack of affordable, high-quality dental care, and mistrust of dentists due to delayed treatment, high cost, and reliance on extraction, as found in other studies [37,38,39]. Notably, families had substantially better access to medical care compared to dental care, as evidenced by most mothers having prenatal care and most children being immunized. The medical system appeared oriented toward preventive care while the dental system was oriented toward urgent treatment for advanced disease. This need for greater public oral health resources and orientation toward preventive services has been identified in many LMICs [39,40,41].

### 4.3. Severe Caries and Undernutrition

Half of the children in our sample had untreated caries, about 1 in 5 children had deep decay in up to 15 teeth, and over 1 in 4 children suffered from mouth pain. Most children with deep caries and mouth pain were ages 3 to 6 years, likely because of the cumulative effects of cariogenic exposures and the decay process over time.

Over half of children in our sample were undernourished, with higher rates of stunting in younger children, and wasting and underweight in older children. For children aged 3 through 6, linear regression analysis demonstrated that for each tooth with deep decay, a child had an additional 10% increased odds of undernutrition. This highlights the opportunity to prevent severe ECC by incorporating oral health promotion, fluoride varnish, and caries arrest treatments into well-child medical care in the first three years of life. These interventions could potentially prevent a substantial amount of suffering from chronic oral infection, mouth pain and associated undernutrition [11,12,42,43].

Additionally, in this model, increased frequency of junk food consumption was associated with a lower odds of undernutrition. We believe that this is due to the calorific effect of junk food, which is also contributing to increased rates of obesity in India and other LMICs [44,45]. Although children with high frequency of junk food consumption may appear less undernourished according to weight and height measures, we were unable to assess their risk for undernutrition by micronutrient deficiencies and anemia [46]. While our sample of children was primarily malnourished, their high rates of junk food consumption may put them at risk of developing obesity, type 2 diabetes, and cardiovascular disease as adolescents and adults [47].

### 4.4. Strengths and Limitations

This is the first study to use a mixed-methods approach to examine junk food consumption, oral health, and nutrition status for children less than seven years in Mumbai, India. A strength of this study is a large sample size. Additionally, the qualitative information helped explain the quantitative findings regarding maternal and child behavior, elucidate limitations in interpreting the quantitative data, and offer potential solutions and future interventions. The collaboration with local NGOs contributed to longer-term focus on oral health and nutrition issues.

A limitation of this study was convenience sampling for the quantitative component and purposive sampling and selective recruitment for the qualitative component, which limit the generalizability of our findings to a broader population. Due to time limitations for interviews, we were able to ask about only cariogenic dietary items rather than a comprehensive diet recall. In addition, the survey was subject to mothers’ recall bias, self-reporting bias, and social desirability bias. Mothers may have under-reported children’s junk food consumption and may not have captured the snacks children consumed independently, or given by other family members. Lastly, with the cross-sectional study design of the quantitative study, we are unable to establish temporality, and thus cannot determine causality.

### 4.5. Recommendations for the Future

This study demonstrates opportunities for improving children’s oral health and nutrition. Education on diet and oral hygiene, as well as dental preventive and treatment services, should be incorporated into primary care maternal and child health services, from prenatal care through childhood [33]. Interventions should engage mothers and other key decision makers in the family (i.e., fathers, mother-in-law, and father-in-law). Educational messages should address long-term adverse effects of ultra-processed foods, promote the traditional diets, and highlight how good oral health and nutrition are associated with positive attributes in Indian culture, strength, attractive appearance, intelligence, and economic potential [48,49]. Public health programs should increase access to affordable dental care for low-income populations. This could involve expanding dental services in public health centers, and conducting periodic community dental camps for oral health promotion, dental screening, preventive care and urgent treatment, with vouchers for adults or children to seek additional dental care. Statewide and countrywide policies to limit the marketing of junk food and proximity of junk food to schools, and expand access to affordable healthy snacks and water, are essential. In addition, India could expand its taxes on sugary sweetened beverages and non-nutritious snack foods. Moreover, culturally-sensitive health promotion efforts should focus on shifting the cultural preferences for sweets, snack foods and sugary tea to healthy, nutritious options. 

Future studies should assess longitudinal causal relationships among junk food consumption, ECC, undernutrition, and obesity. Ecological studies should assess the impact of local food and beverage policies on child oral health and nutrition outcomes over time; and intervention studies should be developed to assess the impact of community-based oral health promotion and treatment to prevent ECC and ECC-related undernutrition. 

## 5. Conclusions

This mixed-methods study of children ages 0 through 6 and their families in low-income urban communities in Mumbai, India, found daily consumption of junk food, progressing frequency and severity of untreated ECC from infancy to age six, frequent mouth pain, and association between severe ECC and under-nutrition. The challenges to ensuring good oral health and nutrition included families giving children daily pocket money to buy junk food, parents not enforcing daily tooth brushing, and poor access to affordable and high-quality dental services. These findings highlight the need to incorporate oral health education and services into primary care maternal-child health services, expand policies to limit marketing of junk food, and develop public health messaging to promote traditional Indian diets, oral hygiene, and preventive oral healthcare. Future studies should assess the effectiveness of community-based oral health interventions on improving children’s oral health and nutrition status.

## Figures and Tables

**Figure 1 ijerph-17-08629-f001:**
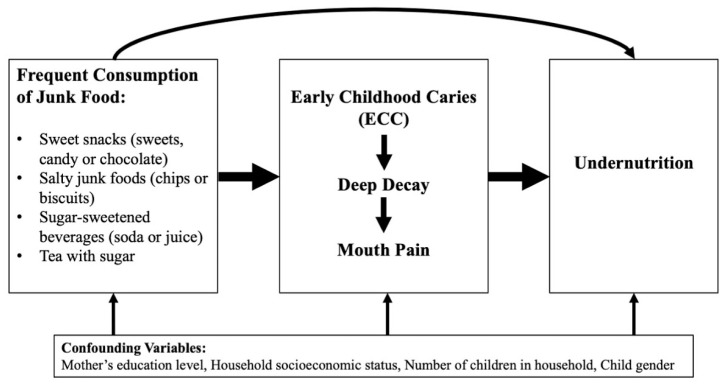
Conceptual model: Pathway from junk food consumption, early childhood caries to undernutrition.

**Figure 2 ijerph-17-08629-f002:**
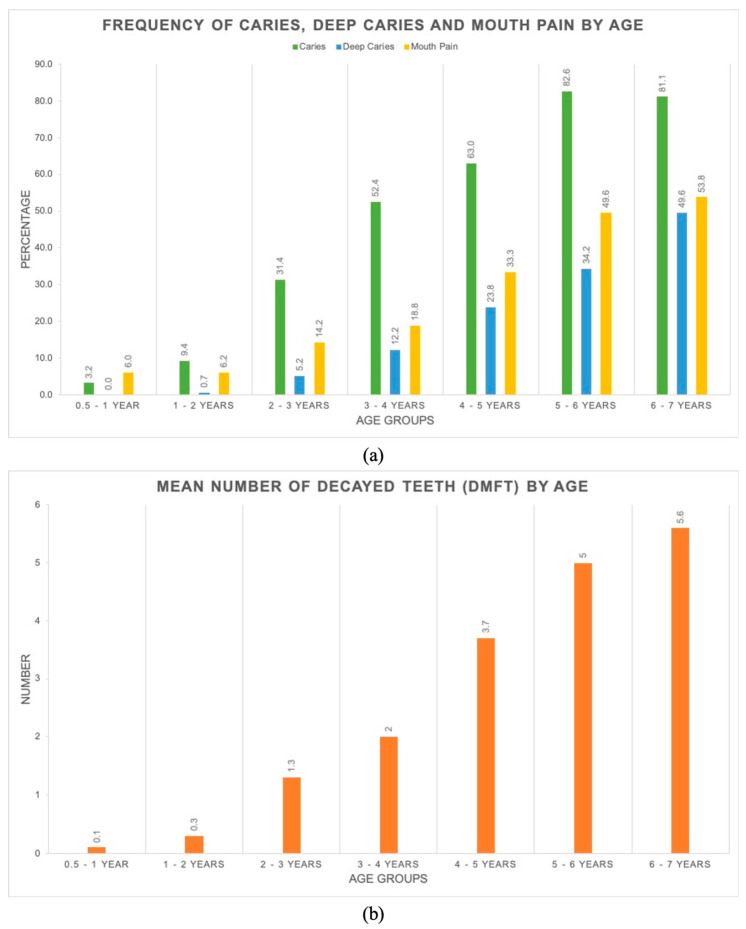
Child caries experience by age (**a**) Frequency of caries, deep caries and mouth pain by age; (**b**) Mean decayed, missing or filled teeth (dmft) by age.

**Table 1 ijerph-17-08629-t001:** Family demographics.

Family Characteristics	Total (*n* = 959 Children and 772 Mothers) *
	Mean ± SD or Number (%)
Mean mother’s age (years)	27.3 ± 4.8
Mean years of mother’s education	6 ± 4.0
Mean number of children	2.4 ± 1.1
Mean household size (number of individuals)	6 ± 2.4
Has electricity at home	902 (97.5)
Has potable water at home	722 (78.1)
Uses cooking fuel other than wood (gas or electric)	753 (81.5)
Child gender	
Female	519 (54.4)
Male	436 (45.7)
Mean child age (years)	3.7 ± 1.8
Child age	
6 months to <1 year	62 (6.5)
1 year to <2 years	139 (14.5)
2 year to <3 years	153 (16.0)
3 year to <4 years	164 (17.1)
4 year to <5 years	181 (18.9)
5 year to <6 years	149 (15.5)
6 year to <7 years	111 (11.6)
Time to walk from home to a store with junk food	
Less than 5 min	630 (80.5)
6–20 min	112 (14.3)
Over 20 min	41 (5.2)

* Excludes missing values; SD: standard deviation.

**Table 2 ijerph-17-08629-t002:** Maternal Oral Health and Nutrition Knowledge, Practices, and Status.

Maternal Characteristics	Total (*n* = 772 Mothers) *
	Mean ± SD or Number (%)
Maternal knowledge on caries risk	
Eating sweets causes caries	695 (75.2)
Not brushing causes caries	281 (30.4)
Drinking soda/juice causes caries	64 (6.9)
Bottle-feeding causes caries	36 (3.9)
Maternal dietary practices	
Daily consumption of	
Milk	188 (20.7)
Soda	30 (3.3)
Chips, biscuits	136 (17.5)
Sweets, candy, chocolate	73 (11.0)
Tea with sugar	578 (86.9)
Maternal oral health practices	
Has her own toothbrush	701 (93.8)
Has been to the dentist	231 (42.3)
Received prenatal care	873 (93.6)
Mean number of prenatal visits	7 ± 4.5
Maternal oral health status	
Symptoms in the past 3 months	
Mouth pain or sensitivity	237 (30.9)
Decayed or loose tooth	66 (8.6)
Bleeding gums	25 (3.3)
Inflammation of the mouth	15 (2.0)
Any dental problems	299 (38.9)

* Excludes missing values; SD: Standard Deviation.

**Table 3 ijerph-17-08629-t003:** Child Dietary and Oral Health Practices.

Child Characteristics	Total (*n* = 959 Children) *	Age < 3	Age ≥ 3
	Mean ± SD or Number (%)	Mean ± SD or Number (%)	Mean ± SD or Number (%)
Child dietary practices			
Breastfed	872 (92.9)	334 (94.9)	538 (91.7)
Mean duration of breastfeeding (months)	21 ± 10.7	15.8 ± 7.6	23.2 ±11.0
Bottle-fed	265 (28.2)	114 (33.0)	151 (25.7)
Mean duration of bottle- feeding (months)	19.4 ± 12.9	14.2 ± 9.9	21.2 ± 13.3
Bottle-fed with sugary drink	20 (4.9)	4 (2.0)	16 (7.6)
Use of bottle during sleep (occasionally/frequently)	115 (45.6)	51 (49.0)	64 (43.2)
Daily consumption of			
Milk	567 (62.8)	241 (72.4)	326 (57.2)
Soda/juice	82 (8.9)	20 (5.9)	62 (10.6)
Sweets, candy, chocolate	489 (52.4)	145 (42.1)	344 (58.5)
Chips, biscuits	543 (58.1)	189 (54.2)	354 (60.4)
Tea with sugar	354 (51.9)	76 (28.8)	278 (66.5)
Money spent on junk food per week			
5–15 Rupees per child (1–2 Rs/day)	170 (26.7)	51 (21.2)	119 (30.1)
15–30 Rupees per child (2–4 Rs/day)	76 (11.9)	27 (11.2)	49 (12.4)
30–50 Rupees per child (5–7 Rs/day)	112 (17.6)	44 (18.3)	68 (17.2)
50–70 Rupees per child (8–10 Rs/day)	114 (17.9)	40 (16.6)	74 (18.7)
Above 70 Rupees per child (>10 Rs/day)	165 (25.9)	79 (32.8)	86 (21.7)
Child oral health practices			
Has his/her own toothbrush	750 (80.7)	203 (59.5)	547 (92.9)
Has toothpaste	763 (91.7)	256 (89.2)	507 (93.0)
Mother helps with brushing (frequently/almost always)	458 (63.2)	150 (70.4)	308 (60.2)
Mother does nothing to care for child’s teeth	267 (36.8)	63 (29.6)	204 (39.8)
Has been to the dentist	103 (14.0)	9 (4)	94 (18.4)
Up-to-date immunizations	887 (97.5)	329 (97.3)	558 (97.6)

* Excludes missing values; SD: Standard Deviation.

**Table 4 ijerph-17-08629-t004:** Child Oral Health and Nutrition Status.

Child Characteristics	Total (*n* = 959 Children) *	Age < 3	Age ≥ 3
	Mean ± SD or Number (%)	Mean ± SD or Number (%)	Mean ± SD or Number (%)
Oral health status			
Frequency of caries	476 (49.6)	63 (17.8)	413 (68.3)
Mean proportion of untreated caries **	0.96 ± 0.15	0.90 ± 0.30	1.00 ± 0.10
Range in number of dmft **	0 to 20	0 to 10	0 to 20
Mean number of dmft ** for all children	2.7 ± 3.9	0.7 ± 1.8	3.9 ± 4.3
Mean number of dmft ** for children with caries	5.4 ± 4.0	3.7 ± 2.5	5.7 ± 4.1
Distribution of number of decayed teeth			
No decayed teeth	483 (50.4)	291 (82.2)	192 (31.7)
1 to 4 decayed teeth	252 (26.3)	43 (12.2)	209 (34.6)
5 to 9 decayed teeth	145 (15.1)	17 (4.8)	128 (21.2)
10 or more decayed teeth	79 (8.2)	3 (0.9)	76 (12.6)
Frequency of deep decay into the pulp	178 (18.6)	9 (2.5)	169 (27.9)
Range in number of deep decay (d3)	0 to 15	0 to 8	0 to 15
Frequency of mouth pain			
Any mouth pain (occasionally/frequently/always)	243 (27.2)	32 (9.8)	211 (37.3)
Mouth pain (frequently/always only)	109 (12.2)	13 (4.0)	96 (17.0)
Mother’s assessment of child’s oral health as “bad”	185 (20.1)	29 (8.7)	156 (26.5)
Mother’s assessment of child’s overall health as “bad”	115 (12.3)	38 (11.0)	77 (13.0)
Nutrition status. HAZ < −2 or WAZ < −2 or BAZ < −2	538 (56.1)	213 (60.3)	325 (53.7)
HAZ < −2	401 (41.8)	163 (46.1)	238 (39.3)
WAZ < −2	342 (35.7)	109 (30.8)	233 (38.5)
BAZ < −2	201 (21.0)	124 (20.5)	77 ± 21.8
Mean Z–score HAZ	−1.6 ± 1.5	−1.7 ± 1.7	−1.6 ± 1.3
Mean Z–score WAZ	−1.6 ± 1.1	−1.5 ± 1.1	−1.6 ± 1.1
Mean Z–score BAZ	−0.8 ± 1.4	−0.7 ± 1.6	−0.9 ± 1.2

* Excludes missing values; SD: Standard Deviation; HAZ: height-for-age Z–score; WAZ: weight for-age Z–score; BAZ: BMI-for-age Z–score; ** d/dmft; dmft: decayed teeth/decayed, missing, or filled teeth

**Table 5 ijerph-17-08629-t005:** Association of Undernutrition with Deep Decay and Junk Food Consumption ^1^

Child Characteristics	Outcome: HAZ, BAZ, or WAZ			
	cOR	95% CI	aOR	95% CI
Children < 3 years				
Presence of deep decay, d3 (continuous)	0.84	0.61–1.16	0.87	0.63–1.20
Junk Food Tertiles	0.81	0.62–1.06	0.85	0.65–1.12
Children ≥ 3 years				
Presence of deep decay, d3 (continuous)	1.10 *	1.01–1.19 *	1.1 *	1.02–1.21 *
Junk Food Tertiles	0.80 *	0.65–0.98 *	0.80 *	0.65–0.98 *

^1^ Adjusted for Mother’s education level, gender, and junk food tertile index (tertiles); cOR—Crude Odds Ratio; aOR—Adjusted Odds Ratio; CI—confidence interval; HAZ: height-for-age Z-score; WAZ: weight for-age Z-score; BAZ: BMI-for-age Z-score; * *p* < 0.05

**Table 6 ijerph-17-08629-t006:** Focus Group Themes and Subthemes.

Theme	Subthemes	Quotes
**(1) Families’ oral health, knowledge, attitudes, practices, and experiences**	General awareness of the importance of good oral health. Common experience with oral health problems in adults and children. Traditional healthy diet and oral hygiene practices. Oral hygiene information from family, neighbors, schools, doctors, television. Barriers to oral hygiene common for children and adults.	*“We were taught how to brush but then the kids just brush in the most haphazard way … our kids just do it fast, they just clean the front and that’s all.”* *“My daughter is 9 years old, she’s only 14 kg and her teeth hurt so much she sleeps without eating.”* *“My cousin sister’s kids teeth are really bad, they’re all bad, none of them are good and he keeps falling ill.”* *“[Purchasing a toothbrush] depends on the house, usually it is budget specific some find it easy some find it difficult.”*
**(2) Contributors to child tooth decay**	Widespread availability of low-cost sugary snacks and drinks. Parents’ busy work/life balancing many priorities–limited time to prepare food and supervise children. Community culture of giving children pocket money to buy snacks. Extended family members spoil children with junk food. Difficulty brushing children’s teeth. Limited access to dental services.	*“My community is filled with shops and all that so kids usually when they wake up, they get hungry, get one or two rupees and go and buy some food. This happens everyday nobody cares about brushing.”* *“We have sweets, tea, cold drinks and snacks on a daily basis … [They] keep our mind fresh, reduce headache and sleep. Tea/Chai is our favorite and we have it up to 2–3 times a day.”* *“Usually my in-laws take them outside to eat secretly to eat chocolate and cake … His weight increased …”* *“I don’t make my son brush because it hurts, and he doesn’t like to brush so I don’t force him. It’s not that it hurts too much, he just doesn’t enjoy it …”* *“Dentists take us only when our teeth are so bad that they can extract it, but they take a lot of money and they take the appointments really late.”*
**(3) Suggestions to improve children’s oral health**	Limits to marketing of junk food to children. Family education on junk food, tooth decay and pocket money. Oral hygiene education and products for toothbrushing. Child education on oral health. Access to affordable, high-quality dental care.	*“We used to try to explain with love but then I spoke and they listened to me. I had to explain really seriously. We had to explain the effect of eating bad food outside with the resultant effect of falling sick and not eating nutritious food.”* *“Give him the habit of brushing when he’s a kid. I have problems, and have had these problems for a while but not with my kids. I gave them the habit since they were kids, to brush.”* *“Whichever parents come into the clinic, their kids are improving so much. The others aren’t improving.”*

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
