# Peer review of "Early Childhood Junk Food Consumption, Severe Dental Caries, and Undernutrition: A Mixed-Methods Study from Mumbai, India"

_ijerph, 2020, doi:10.3390/ijerph17228629_

Round 1
Reviewer 1 Report
English should be revised. Sometime there are very long sentences that make the paper difficult to read and understand.
The acronym NGO is not defined in the text.
Was the study approved by an ethics committee? This should be reported.
There must be a phrase about respecting the Helsinki declaration.
There must be an indication of the inclusion and exclusion criteria
Most of the information in the tables is repeated in the text. This should not happen.
In the tables the units are not correct for all lines (N and% / mean and SD) - the reading of the tables becomes confusing.
There is a great repetition of results throughout the discussion. This should be avoided as much as possible and the discussion should be more concise.
It could be interesting to discuss possible public health measures to be applied in the country, considering their possibilities.
Author Response
Point 1: English should be revised. Sometimes there are very long sentences that make the paper difficult to read and understand.
Response 1: Thank you for your comments. We have made changes in the text to address the longer sentences obstructing the clarity of the paper. There are 9 instances in which we’ve either: cut a longer sentence into two small concepts or refined wording and removed words from a long sentence. These changes span the Introduction, Methods and Discussion sections. Please see lines 58-63, 119-121, 131-132, 427-428, 438-439, 456-459, 464-466.
Point 2: The acronym NGO is not defined in the text.
Response 2: Thank you for this comment. We have defined an NGO as a non-governmental organization the first time it’s mentioned in the text. Please see lines 106-107.
Point 3: Was the study approved by an ethics committee? This should be reported.
Response 3: Thank you for this comment and bringing this important point to our attention. It appears that our initial paragraph in the Methods section, entitled “Study Design and Population,” which included the names of the partner organizations and details of ethics committee approval, was inadvertently omitted from our draft manuscript. We have re-inserted this paragraph Please see lines 96-112.
Point 4: There must be a phrase about respecting the Helsinki declaration.
Response 4: Thank you for this comment. We have added information respecting the Helsinki Declaration on Lines 111-112.
Point 5: There must be an indication of the inclusion and exclusion criteria
Response 5: Thank you for this comment. We included all families in this study with children between 6 months through 6 years, with no exclusions for medical reasons or otherwise. Please see the inclusion and exclusion criteria specified in lines 101-102.
Point 6: Most of the information in the tables is repeated in the text. This should not happen.
Response 6: Thank you for this comment. We have reviewed the tables and Results section and have made 18 changes after carefully removing extra details from the text that are repeated in Tables 1-3. These changes involve deleting frequencies (already reported in the Tables) from the text to reduce repetition.
Point 7: In the tables the units are not correct for all lines (N and% / mean and SD) - the reading of the tables becomes confusing.
Response 7: Thank you for this feedback. We have made significant changes to the formatting of the tables to prevent further confusion. To consolidate the label for N/% and mean/SD we have revised the subheading to “Mean ± SD or No. (%)”. We have revised the presentation of the data to match the subheading as mentioned in Tables 1-4 with major changes noted in “Track Changes” on the final document. We hope these changes make the table and units more clear.
Point 8: There is a great repetition of results throughout the discussion. This should be avoided as much as possible and the discussion should be more concise.
Response 8: Thank you for this comment. There are over 3 places where we have removed results presented in the Discussion to prevent redundancies. There are some results, however, which we have kept in the Discussion to highlight how our study findings relate to those of other studies, as well as implications and recommendations. Please see Lines 438-439, 463-466, 507-510.
Point 9: It could be interesting to discuss possible public health measures to be applied in the country, considering their possibilities.
Response 9: Thank you for this comment. We agree that it is important to discuss public health interventions that could be implemented in India. We detailed some recommendations for policies and programs in Section 5 of the Discussion: Recommendations for the Future. Please see Lines 982-991. We also have added a sentence regarding potential policies to expand India’s sugar-sweetened beverage tax– see Lines 998-999.
Reviewer 2 Report
This a very important study which indicates that ECC can be a significant risk factor of under-nutrition in a low-income community in Mumbai. Authors present a lot of factors contributing to ECC in examined children and based on the interviews with selected group of caregivers discuss possible solutions which could help to improve oral health of the community.
Since the main aim was to describe the risk factors for ECC, paying special attention to junk food consumption, I would be curious whether there was any significant association between poor dietary habits and ECC (dental caries is a multifactorial disease). As far as I can see, junk food consumption reduced the risk of under-nutrition, probably due to the caloric effects of junk food. At the same time ECC increased a risk of under-nutrition. The effects of ECC and poor diet are opposite.
I am enclosing the MS with several minor corrections.
Author Response
Point 1: This is a very important study which indicates that ECC can be a significant risk factor of under-nutrition in a low-income community in Mumbai. Authors present a lot of factors contributing to ECC in examining children and based on the interviews with selected groups of caregivers discuss possible solutions which could help to improve oral health of the community. Since the main aim was to describe the risk factors for ECC, paying special attention to junk food consumption, I would be curious whether there was any significant association between poor dietary habits and ECC (dental caries is a multifactorial disease). As far as I can see, junk food consumption reduced the risk of under-nutrition, probably due to the caloric effects of junk food. At the same time ECC increased the risk of under-nutrition. The effects of ECC and poor diet are opposite.
Response 1: Thank you for this comment. We agree that it would be important to examine the associations between poor dietary habits and ECC. From our literature review, we concluded that the association between frequent consumption of sugary drinks and snacks with ECC has been well-established in the global oral health literature over the past several decades; and studies from India and other LMIC have already described the association between junk food consumption and ECC (Patil 2018, Ismail 1998, Naidu 2013). For this study, our hypothesis-driven analysis aimed to build on the well-established association between junk food consumption and ECC to describe junk food consumption in this study population and assess the association between severe ECC and malnutrition. We believe that this is an important focus of this analysis since India continues to have among the greatest number of children with malnutrition worldwide, and ECC has been an under-researched and potentially preventable and treatable contributor to malnutrition. Our future studies will continue to examine the association between children’s frequent consumption of junk food with ECC, and examine the relationships between ECC and nutrition status.
Point 2: I am enclosing the MS with several minor corrections.
Response 2: Thank you for this comment and your corrections. We have made the changes you suggested in the final revised document.
References:
Ismail AI. The role of early dietary habits in dental caries development. Spec Care Dentist. 1998 Jan-Feb;18(1):40-5. doi: 10.1111/j.1754-4505.1998.tb01357.x. PMID: 9791306.
Naidu R, Nunn J, Kelly A. Socio-behavioural factors and early childhood caries: a cross-sectional study of preschool children in central Trinidad. BMC Oral Health. 2013 Jul 9;13:30. doi: 10.1186/1472-6831-13-30. PMID: 23834898; PMCID: PMC3708808.
Patil, S.; Jain, R.; Shivakumar, K.; Srinivasan, S. Sociodemographic and behavioral factors associated with early childhood caries among preschool children of Western Maharashtra. Indian Journal of Dental Research 2018, 29 (5), 568 DOI: 10.4103/ijdr.ijdr_158_17.